# The PACAP/PAC1 Receptor System and Feeding

**DOI:** 10.3390/brainsci12010013

**Published:** 2021-12-23

**Authors:** Keerthana Sureshkumar, Andrea Saenz, Syed M. Ahmad, Kabirullah Lutfy

**Affiliations:** 1UCLA College of Letters and Sciences, University of California, 612 Charles E Young Dr. South, Los Angeles, CA 90095, USA; keerthanasuri77@gmail.com; 2College of Pharmacy, Western University of Health Sciences, 309 East Second Street, Pomona, CA 91766, USA; andrea.saenz@westernu.edu (A.S.); smahmad@westernu.edu (S.M.A.)

**Keywords:** PACAP, PAC1, homeostatic, hedonic, food intake, signaling, neuroanatomical site

## Abstract

Pituitary adenylyl cyclase activating polypeptide (PACAP) belongs to the vasoactive intestinal polypeptide (VIP)/secretin/glucagon superfamily. PACAP is present in two forms (PACAP-38 and PACAP-27) and binds to three guanine-regulatory (G) protein-coupled receptors (PAC1, VPAC1, and VPAC2). PACAP is expressed in the central and peripheral nervous systems, with high PACAP levels found in the hypothalamus, a brain region involved in feeding and energy homeostasis. PAC1 receptors are high-affinity and PACAP-selective receptors, while VPAC1 and VPAC2 receptors show a comparable affinity to PACAP and VIP. PACAP and its receptors are expressed in the central and peripheral nervous systems with moderate to high expression in the hypothalamus, amygdala, and other limbic structures. Consistent with their expression, PACAP is involved in several physiological responses and pathological states. A growing body of literature suggests that PACAP regulates food intake in laboratory animals. However, there is no comprehensive review of the literature on this topic. Thus, the purpose of this article is to review the literature regarding the role of PACAP and its receptors in food intake regulation and to synthesize how PACAP exerts its anorexic effects in different brain regions. To achieve this goal, we searched PubMed and reviewed 68 articles regarding the regulatory action of PACAP on food intake. Here, we present the literature regarding the effect of exogenous PACAP on feeding and the role of endogenous PACAP in this process. We also provide evidence regarding the effect of PACAP on the homeostatic and hedonic aspects of food intake, the neuroanatomical sites where PACAP exerts its regulatory action, which PACAP receptors may be involved, and the role of various signaling pathways and neurotransmitters in hypophagic effects of PACAP.

## 1. Introduction

Food consumption is necessary for survival and is affected by homeostatic and hedonic aspects of feeding. Hunger and satiety are two physiological responses that promote and reduce appetite and food intake. Signals that facilitate or impede appetite and feeding are integrated into the nucleus tractus solitarius and hypothalamus. Three hypothalamic structures play essential roles in this process. These include the ventromedial nucleus of the hypothalamus (VMN), serving to suppress appetite and food intake; the lateral hypothalamus (LH), functioning to promote appetite and food consumption; and the arcuate nucleus of the hypothalamus (ARC), operating as a switchboard to turn on or turn off the above structures to regulate appetite and food intake [1,2]. 

PACAP mediates food intake in these structures in distinctive manners. For example, PACAP injection in the ARC selectively decreases carbohydrate intake but not high-fat intake. The role of afferent projections from the bed nucleus of stria terminalis (BNST) to the LH in homeostatic feeding is to attenuate food intake upon PACAP injection, suggesting that PACAP may also have indirect effects on the LH’s satiety-dependent food intake (for detail, see Section 4 and Section 8).

Two groups of neurons exist in the ARC that integrate and modulate feeding behaviors. These neurons contain anorexigenic peptides, such as proopiomelanocortin (POMC) and cocaine-amphetamine-regulated transcript (CART), as well as orexigenic peptides, such as neuropeptide Y (NPY)/agouti-related peptide (AgRP). Many central and peripheral inputs can alter the activity of these neurons. For example, insulin, leptin, ghrelin, etc., secreted by peripheral tissues, can activate one or the other type of ARC neurons to increase or decrease appetite and food intake. 

Changes in the levels of nutrients in blood circulation can also relay excitatory or inhibitory signals to these neurons to stimulate or suppress appetite. For example, low blood glucose increases ghrelin secretion, which increases calcium levels in NPY-containing neurons [3]. The increase in calcium facilitates exocytosis and triggers the release of NPY, promoting appetite and food intake.

Several peptides in the peripheral or central nervous system can regulate appetite and food intake. The glucagon superfamily promotes hypophagia, a conserved response throughout different species [4]. A growing body of evidence suggests that pituitary adenylyl cyclase activating polypeptide (PACAP), a peptide within this superfamily, regulates food intake. Thus, PACAP, like other analogs in the glucagon superfamily, may prove to be a potential target to develop medications to treat obesity. Likewise, the dual role of PACAP in both satiety and hedonic feeding suggests that it may prove useful in treating binge-eating disorders. In this article, we reviewed the literature (68 articles selected and included in this review) regarding the regulatory action of PACAP on food intake (Table 1).

In this review, we attempted to describe the literature on the role of endogenous PACAP and the effect of exogenous PACAP on food intake. Furthermore, we attempted to provide information regarding neuroanatomical sites where PACAP exerts its hypophagic effects in the brain. We also discussed the effect of PACAP on the hedonic and homeostatic aspects of feeding. Finally, we reviewed the role of neurotransmitters and signaling mechanisms in the hypophagic actions of PACAP.

## 2. PACAP, Its Receptors, and Its Physiological Roles

Pituitary adenylate cyclase-activating polypeptide (PACAP) was initially isolated from the ovine hypothalamus as an activator of adenylyl cyclase in pituitary cells [5]. PACAP is found in two forms. PACAP38 comprises 38 amino-acid residues and PACAP27, a C-terminally alpha-amidated form of PACAP; the sequence encompasses an internal cleavage-amidation site that generates a 27-residue alpha-amidated polypeptide fragment or PACAP27, which corresponds to the N-terminal 27 amino-acid portion of the PACAP38 [6].

Human PACAP27 shares 68% identity with VIP, classifying PACAP as a member of the VIP/secretin/growth hormone/glucagon superfamily. The PACAP gene contains five exons, and PACAP is encoded by exon 5. The human PACAP promoter possesses two cAMP response-like elements, which play a role in the tissue-specific factor growth hormone and six binding domains for the thyroid-specific transcription factor-1.

PACAP binds to three G protein-coupled receptors: PAC1R, VPAC1, and VPAC2 [7]. The PACAP-preferring PAC1 receptors isoforms can be coupled to G_s_ and G_q_ to stimulate adenylyl cyclase and phospholipase C (PLC), respectively. In contrast, VPAC1 and VPAC2 receptors, which had an equal affinity for PACAP and VIP, are principally coupled to G_s_ to activate adenylyl cyclase and increase intracellular 5′-cyclic adenosine monophosphate (cAMP) levels [8]. 

The rise in cAMP activates protein kinase A (PKA). PLC causes the breakdown of phosphatidylinositol and the production of inositol triphosphate and diacylglycerol, leading to an increase in intracellular calcium and protein kinase C activation. These biochemical changes modulate multiple ionic currents and stimulate signaling factors. Moreover, PAC1 receptor activation led to β-arrestin-mediated receptor internalization and endosomal signaling, which led to extracellular regulated protein kinase phosphorylation [9,10].

PACAP is expressed in the central nervous system and peripheral organs and exerts multiple effects (pleiotropic). These include the control of neurotransmitter release, vasodilation, bronchodilation, activation of intestinal motility, increase in insulin and histamine secretion, immune modulation, as well as cell proliferation and differentiation [6]. A growing body of literature also suggests that PACAP and its receptors are involved in homeostatic and hedonic aspects of feeding. PACAP exerts its anorexigenic effect in different brain regions and periphery involving insulin regulation and other mechanisms. This article aims to evaluate the literature regarding the regulatory actions of PACAP on feeding.

## 3. PACAP and Feeding

Initial studies showed that central PACAP administration reduces feeding. Morley and colleagues found that intracerebroventricular (ICV) injection of PACAP at 2.5 (but not 1) µg reduced food intake in mice [11]. Likewise, a dose-dependent decrease in food intake was reported following ICV injection of PACAP in male Sprague–Dawley rats [12]. Subsequent studies confirmed these results showing that ICV administration of PACAP to food-deprived male CD1 mice dose-dependently reduced food consumption. Matsuda and colleagues [13,14] found that ICV injection of PACAP or VIP (11 and 22 pmol/g) reduced food consumption in goldfish, with the higher dose of PACAP and VIP inducing increases in locomotor activity as well. 

In another study using Zebrafish, ICV injection of a low dose of PACAP (2 pmol/g) suppressed food intake, suggesting that PACAP acts as an anorexigenic peptide in the brain of fishes [7]. The hypophagic effect of PACAP was reduced in the presence of a PACAP receptor antagonist [15] or PACAP receptor knockout mice [16]. Additional research showed that the peptide levels were elevated in the brainstem in food-deprived rats. 

Similar changes occurred in the hypothalamus and telencephalon in chickens [17], suggesting that endogenous PACAP is involved in food intake regulation. In this article, we reviewed the literature regarding the role of PACAP and its receptors in food intake. First, we discussed the role of endogenous PACAP in feeding. We next presented the findings regarding the effect of exogenous PACAP and related ligands on food intake. We finally reviewed the literature regarding the involvement of different neurotransmitters and brain regions in the hypophagic effects of PACAP.

### 3.1. The Role of Endogenous PACAP in Food Intake

Fasting alters the level of endogenous PACAP. Jozsa and colleagues [17] reported elevated levels of PACAP in the brainstem in food-deprived rats and the hypothalamus and telencephalon in chickens, raising the possibility that endogenous PACAP may serve as a regulator of food intake. Similar results were reported in mice. Kiss et al. (2006) found that PACAP levels were changed in mice after food and water deprivation, and there was sexual dimorphism in this regard. In the hypothalamus, brainstem, and telencephalon of male mice, PACAP levels were significantly elevated 12 h after the beginning of food deprivation, but not at 36 or 48 h [18]. 

No significant differences were observed in PACAP levels between the control and experimental conditions in the brainstem and diencephalon in female mice. In the hypothalamus of food-deprived female mice, PACAP levels showed a slight increase. When male mice were water-deprived, there was no significant change in PACAP levels, but PACAP levels were decreased in the hypothalamus of female mice at the 12 and 36 h time points. The brainstem of water-deprived male mice showed slight increases in PACAP levels and returned to basal levels by 48 h. 

PACAP levels in female mice displayed significant decreases in the brainstem 12 h after the onset of water deprivation. These results suggest that there is sexual dimorphism in the expression of PACAP in response to starvation, which may explain male/female differences in the food intake reported in the literature. There were no significant changes in PACAP levels in the telencephalon of either male or female mice. However, whether these changes are due to the stress of fasting—since PACAP was reported to regulate the stress response [19,20,21,22,23,24,25]—or whether these changes promote satiety in animals undergoing food-deprived need to be addressed.

PACAP levels in the VMN of mice [26] and goldfish [13] are upregulated by chronic overeating. On the other hand, its expression is reduced by food deprivation in both the whole hypothalamus and the VMN of the mouse. Moreover, in the rat hypothalamus, PACAP protein levels are increased 12 h after fasting but returning to basal levels by 36 h. Gargiulo and colleagues concluded that PACAP regulates food intake through the negative feedback loop under normal homeostatic feeding; this may become dysregulated under food addiction [reviewed in [27]].

Nakata and colleagues showed that food intake was reduced in PACAP-deficient mice compared to their wild-type controls, showing that endogenous PACAP may facilitate eating rather than promoting satiety [16]. They also examined the underlying mechanism of this regulatory action of PACAP and discovered that NPY mRNA was reduced in PACAP-deficient mice. 

Given that NPY neurons are affected by glucose and other chemicals, they also studied if high carbohydrate intake would be changed in these mice. Their results revealed that carbohydrate but not high-fat consumption was reduced in PACAP knockout mice than wild-type controls [16]. Together, these findings suggest that endogenous PACAP influences carbohydrate consumption via the NPY-releasing neuron in the ARC. Relatedly, PACAP, PAC1R, and VPAC2R are expressed on NPY-containing neurons in the ARC [28].

Consistent with the above findings, Sherwood [29] found that when PACAP knockout and wild-type mice were fed either a regular-chow or high-fat diet for 30 weeks, there was no difference in growth, food intake, body weight, or size of perirenal fat pads between mice of the two genotypes. However, Nguyen and colleagues [6] showed that the nocturnal and daily food intake was reduced in PACAP knockouts compared to wild-type mice; however, PACAP knockouts showed an increase in food intake during the day, suggesting that differences in food intake in mice lacking PACAP may be dependent on the time of day.

The PACAP system may also be involved in glucose homeostasis. Gray and colleagues reported that glycogen levels were reduced in mice lacking PACAP than their wild-type controls [30]. These researchers discovered that the fasting glucose level was lower with a concomitant increase in insulin levels in PACAP-deficient mice than their wild-type controls [30]. Furthermore, there was an increase in cholesterol, triglycerides, and free fatty acid (although the latter was insignificant) in PACAP knockout mice. These results show that the metabolism of carbohydrates and lipids is impaired in the absence of PACAP [30].

The endogenous PACAP may also regulate blood glucose. Green and colleagues (2006) found that blood glucose levels were significantly elevated at 30 and 60 min after a glucose challenge in mice injected with PACAP6-27, a PACAP receptor antagonist, compared with mice treated with saline. The authors concluded that acute inhibition of PACAP signaling leads to poor glucose tolerance. These authors also showed that once-daily injection of PACAP6-27 for two weeks led to impaired glucose tolerance, the exaggerated glycemic response to feeding with no change in insulin release. Plasma glucagon, triglycerides, total cholesterol, HDL levels were unchanged [31], suggesting a decrease in insulin receptor sensitivity or signaling mechanism. When insulin was administered intraperitoneally, knockout mice fed a high-fat diet but not regular chow showed decreased glucose levels than wild-type mice, suggesting that knockouts were more insulin-sensitive than wild-type controls [29].

Some studies revealed that mice lacking the PACAP gene showed lower food intake levels. These mice also had reduced NPY expression in the ARC, explaining the reduced high-carbohydrate consumption. Contrastingly, other studies reported that when rats were administered PACAP-38 locally in the nucleus accumbens, they showed a reduction in binge-eating and hedonically driven consumption of fats and carbohydrates [32].

Endogenous PACAP may also regulate the release of other peptides, such as insulin and leptin, involved in energy homeostasis. Indeed, the peptide levels were reduced in PAC1 knockout mice under a fasting state and postprandially [15]. The level of GLP-1 was also reduced but only postprandially, whereas the level of PYY and glucagon was unchanged in PAC1 knockout mice. These findings suggest that the PACAP/PAC1 receptor system could be a potential target to develop medications to treat obesity and diabetes.

Researchers proposed that chronic exposure to solutions induces changes in the expression of the splice variants of the PAC1 receptor and thus changes in PAC1R binding and signaling. In areas of the brain in which both PAC1 and VPAC receptors were present, this change of the variant of PAC1 receptor may favor signaling towards either type of receptor, leading to differences in behaviors (Ref. [33]; for an excellent review, see [27]).

Burgos and colleagues measured changes in food intake and body weight induced by cocaine/amphetamine-regulated transcript (CART) in the presence of subcutaneous injection of PACAP6-38 [34]. While PACAP6-38 reduced the ability of CART to decrease food intake and body weight, it did not alter the motor stimulatory effect of CART. PACAP6-38 also had no effects by itself on feeding after peripheral administration. The authors concluded that there might be multiple CART receptors. Alternatively, CART may be acting via the release of endogenous PACAP to reduce food intake. PACAP6-38 might act as an agonist on the VPAC receptors to alter the action of CART on food intake but not on locomotion.

Recently, Hannibal and colleagues [35] studied the role of PACAP in food anticipatory activity (FAA). Researchers found that when PAC1 knockouts and wild-type mice entrained to both the full photoperiod (FPP) and skeleton photoperiod (SPP) were placed on a restricted feeding diet, knockout mice showed an earlier onset of FAA, greater weight loss, and increased activity than their wild-type counterparts at both photoperiod conditions at the 10-lux light intensity but only at the SPP condition at the 300-lux light intensity [35].

In summary, these findings suggest that endogenous PACAP is regulated by food deprivation and overeating, and endogenous PACAP reduces food intake in fishes, rodents, and chickens. Likewise, endogenous PACAP may be involved in the regulatory action of CART and other neuropeptides regulating food and energy homeostasis. Endogenous PACAP has also been shown to regulate ghrelin levels. For example, the ghrelin level is significantly elevated by overnight fasting and postprandially in PAC1 receptor knockout mice compared to their wild-type controls.

### 3.2. The Effect of Exogenous PACAP and Related Ligands on Food Intake

Preclinical studies reported that exogenous PACAP and related ligands mimic the regulatory action of endogenous PACAP on feeding. Here, we present the results of some studies in chronological order. Morley and colleagues found that ICV injection of PACAP at 2.5 (but not 1) µg reduced food intake in mice [11]. Subsequent studies in rats showed a dose-dependent decrease in food intake following ICV injection of PACAP in male Sprague–Dawley rats. The reduced food intake was observed for 16 h following ICV injection; the effect was evident also at 24 and 48 h but returned to control level at 72 h after PACAP injection. 

Additionally, bodyweight decreased, and the rats were more energetic after PACAP administration. PACAP also increased mRNA levels of NPY in the ARC and galanin in the PVN. On the other hand, it decreased CRH levels in the PVN. Rats deprived for 72 h expressed greater NPY mRNA compared to fed rats. CRH mRNA was lower, and galanin mRNA was not affected by fasting. Overall, these results suggest that PACAP exerts hypophagia, but this effect may be independent of changes in the expression of hypothalamic peptides involved in food intake [12].

Matsuda and colleagues found that ICV injection of PACAP or VIP (11 and 22 pmol/g) reduced food consumption in goldfish, with the higher dose of PACAP and VIP inducing increases in locomotor activity as well [14]. In another study using Zebrafish, ICV injection of a low dose of PACAP (2 pmol/g) suppressed food intake, suggesting that PACAP acts as an anorexigenic peptide in the brain of fishes [7]. However, certain synthetic forms of PACAP did not display hypophagic effects at a 10-fold higher dose, indicating that the peptide is anorexigenic only at low doses or that synthetic peptides may desensitize the receptor at higher doses. 

The authors also found that locomotor activity was unchanged before and after administering synthetic/natural peptides. This result suggests that PACAP-induced hypophagia was not a result of motor impairments. However, we found that ICV injection of higher doses of PACAP reduced locomotor activity in C57BL/6J mice [36]. Thus, the difference could be due to different doses of the peptide vs. synthetic ligands or the use of different species of animals.

ICV PACAP administration to food-deprived male CD1 mice dose-dependently reduced food consumption for at least 3 h. This effect of PACAP was attenuated by PACAP6-38, a PACAP receptor antagonist, but was not mimicked by VIP, suggesting that PACAP exerts its anorexigenic action through PACAP-selective PAC1 receptors [37]. However, additional studies are needed to delineate the role of each PACAP receptor in the hypophagic effect of the peptide because PACAP6-38 was shown to possess actions on other PACAP receptors [38,39].

Systemic PACAP administration also reduces feeding. For instance, intraperitoneal injection of PACAP38 and PACAP27 before the dark cycle in mice reduced food intake, meal duration, meal frequency, and feeding bouts. The regulatory actions of these peptides were absent in mice lacking PAC1 receptors, suggesting that PACAP reduces feeding via the PAC1 receptors [15]. These authors also showed that the regulatory actions of PACAP might be through decreases in plasma ghrelin levels, with no change in the glucagon-like peptide 1 (GLP-1), glucagon, PYY, and insulin levels. As stated above, the ghrelin level was significantly elevated by overnight fasting and postprandially in PAC1 receptor knockout mice compared to their wild-type controls. These results suggest that PACAP may reduce food intake by inhibiting the action of ghrelin in the hypothalamus. Alternatively, PACAP may serve as a regulator of ghrelin secretion at the periphery.

The effect of PACAP on blood glucose in Goto Kakizaki (GK) rats and C57BL/6J mice fed a regular versus high-fat diet was studied [40]. The authors found that PACAP38 (0.6 and 60 pmol/kg/day, i.p.) administration at three weeks of age until 8 to 10 weeks reduced blood glucose in GK rats, an animal model of type 2 diabetes. They also assessed the effect of PACAP on high-fat diet-induced type 2 diabetes in C57BL/6J mice. 

The results showed PACAP27 (50 pmol/kg/day for five days) reduced blood glucose in C57BL/6J mice regardless of being fed a regular or high-fat diet. These studies suggest that PACAP reduces glucose in animal models of type 2 diabetes. However, more work is needed to describe how PACAP regulates glucose homeostasis. For example, it needs to be determined whether PACAP exerts its action on insulin secretion or other mechanisms may be involved [40].

Maxadilan, a PAC1R agonist, also reduced food intake in NIH mice [41]. However, M65, the PAC1R antagonist, did not alter food intake in these mice. These authors demonstrated that maxadilan increased food intake upon repeated administration, requiring further investigation to delineate the underlying mechanisms of acute vs. chronic effects of PACAP ligands on food intake and body weight.

The effect of systemic administration of PACAP was evaluated on food intake in rodents. For example, Vu and colleagues [15] examined the effect of PACAP on appetite and food intake. These authors injected PACAP38 or PACAP27 intraperitoneally in PAC1 receptor-deficient mice and their wild-type controls. They found that PACAP, in the periphery, can suppress appetite in a dose-dependent manner in wild-type but not in PAC1R knockout mice, showing that PACAP reduces food intake via the PAC1R in the periphery. The authors also showed that PACAP reduced ghrelin levels without changing the GLP-1, PYY, insulin, and glucagon levels. 

Correspondingly, the ghrelin level increased to a greater extent in knockout animals than their wild-type controls regardless of whether animals were kept fasted or not. These results suggest that exogenous PACAP and endogenous PACAP regulate the level of ghrelin. Interestingly, the levels of other peptides, such as insulin, leptin, and GLP-1, were reduced in PAC1R (−/−) mice. However, in another study, PACAP27 (10–100 fmol) altered insulin secretion from the pancreatic beta cells, which was glucose-dependent and mediated by an increase in calcium concentration in beta cells [42]. 

PACAP27 stimulated insulin secretion and increased calcium concentration inside the beta cells in the presence of 8.3 mM but not 2.8 mM glucose. PACAP38 also increased the release of insulin [42]. The effect of PACAP was blocked by nitrendipine showing the calcium influx via the L-type calcium channels in insulin exocytosis. VIP also increased insulin secretion but at a higher concentration. PACAP38 mimicked the effect of PACAP27. The authors concluded that PACAP could act as a neuronal or local hormonal regulator of glucose-induced insulin secretion [42].

In another study, PACAP was shown to promote glucose output [43]. For example, these authors showed that PACAP27 concentration-dependently increased the glucose output from the rat liver in a calcium-dependent manner. The same effect was observed with PACAP38, whereas VIP only slightly increased the glucose output. Correspondingly, there was an increase in cAMP levels by PACAP27 and PACAP38. When theophylline was added to the perfusate, the cAMP output induced by 40 nM PACAP27 increased significantly; however, the degree of glucose output was not affected.

On the other hand, when the effect of PACAP on glucose output and cAMP accumulation was studied in a calcium-free medium, the stimulatory effect of PACAP on glucose output was blunted without any effect on cAMP accumulation [43]. Together, these results suggest that PACAP may elevate glucose output by increasing calcium but not cAMP, suggesting it may involve the PKC pathway but not the PKA pathway. However, further studies are needed to confirm the involvement of these pathways in the ability of PACAP to increase the hepatic glucose output.

Intraperitoneal PACAP injection increased body weight (without altering hepatosomatic index) via growth hormone through protein increase and lipid mobilization during ovulation in juvenile tilapia and catfish. Injection of VIP at a higher dose (88 pmol/g, i.p.) did not attenuate food consumption, indicating that VIP at a high dose may desensitize the pathways involved in food intake [44]. Matsuda and researchers (2005) found that injection of VIP (22 and 44 pmol/g, i.p.) decreased food intake, while 88 pmol/g of VIP did not have this effect. On the other hand, PACAP decreased food consumption at all doses administered (22, 44, and 88 pmol/g, i.p.).

## 4. Neuroanatomical Sites of PACAP Action in Regulating Food Intake

Different brain regions are implicated in the regulatory actions of PACAP on the homeostatic and hedonic aspects of food intake (Figure 1).

The bed nucleus of stria terminalis (BNST) is important in regulating anxiety-like behavior caused by long-term threats. It also has a role in mediating stress-induced anorexia and associated reductions in weight gain [45,46]. PACAP may act in the BNST to regulate stress response during stressful events. An earlier study [47] showed that infusion of PACAP38 in the posterior BNST in male and female Sprague–Dawley rats mimics weight loss and anorectic state induced by repeated stress exposure. However, this effect was not observed following PACAP administration in the anterior BNST or ICV, which shows that PACAP exerts its action in the posterior BNST. The authors concluded that, in female rats, the site-specific action of PACAP in the BNST was inconclusive. Given the role of PACAP and the importance of BNST in the regulation of food intake and stress response, the PACAP/PAC1R system may be involved in stress-related eating disorders.

In an earlier study [48], PACAP injected directly into the VMN increased both core body temperature and locomotor activity for up to 7 h. When PACAP (25 pmol) was administered locally into the VMN, feeding decreased by 52%, which lasted for 3 h, while PACAP at 50 pmol reduced feeding by 80% for up to 6 h. The study also found that, when PACAP was injected into the VMN, POMC mRNA expression was greater in the ARC, without any change in NPY or AgRP mRNA activity [48]. Considering that the VMN was shown to project onto the nucleus tractus solitarius (NTS), which is responsible for regulating energy homeostasis, food intake, and locomotor activity, researchers concluded that excitation of the VMN via PACAP injections might be responsible for increased POMC signaling within the NTS [48].

When PACAP was injected into the PVN or VMN, it reduced the latency in consuming the first meal and the eating rate in Sprague–Dawley rats compared to its vehicle. When injected in the PVN, rats showed strong decreases in meal size, meal duration, and the total time spent eating. These effects were reversed by PAC1R antagonists but not VPAC antagonists. Researchers also found that PACAP administration in both the PVN and VMN increased the plasma glucose concentration but did not affect pancreatic hormone levels. Upon the injection of PACAP in the VMN, the core body temperature and spontaneous locomotor activity in rats were elevated compared to the controls. These results were reproduced using maxadilan, a PAC1R agonist, but not after VIP injections in either brain region [49].

Iemolo and colleagues found that rats infused with PACAP in the central amygdala showed decreased food intake and body weight with no significant locomotor activity changes [50]. However, intra-basolateral amygdala infusion of PACAP was without any effects on food intake and body weight or locomotor activity, highlighting the site-specificity of hypophagic action of PACAP.

In summary, PACAP acts in different brain regions to regulate food intake. It exerts its action on homeostatic feeding in the hypothalamus, whereas its action in the VTA and NAc may inhibit hedonic feeding (see below). However, further studies are needed to delineate the role of other brain regions, such as the paraventricular nucleus of the thalamus (PVT), which contains high levels of PACAP27 and has been implicated in alcohol self-administration [51,52,53].

## 5. The Role of Other Neurotransmitters in the Actions of PACAP on Food Intake

The regulatory effect of PACAP on food intake may be mediated via the α-melanocyte-stimulating hormones (α-MSH) system [37]. These authors showed that food deprivation increased the expression of NPY and reduced the expression of proopiomelanocortin (POMC), the precursor of the anorexigenic peptide α-MSH. Similar changes were observed in the PACAP gene expression [37], indicating that PACAP may serve as another anorexigenic peptide. Indeed, these authors showed that exogenous PACAP administration reduced food intake in food-deprived animals. 

This effect of PACAP was reduced but not completely abolished by PACAP6-38, a PACAP receptor antagonist. The antagonist alone did not affect food intake. The authors concluded that PACAP acts on PAC1 receptors to reduce food intake. Although this conclusion may be plausible, considering that PACAP acts on VPAC1 and VPAC2 receptors and that PACAP6-38 has some actions at VPAC receptors [38,39], further studies are needed to characterize the receptor types involved in this process.

As stated above, anorexia induced by central injection of PACAP is mediated, in part, through activation of the α-MSH, as the POMC gene expression was increased by PACAP and reduced in PAC1 receptor knockout mice. Additionally, PACAP increased c-fos expression in POMC neurons. Furthermore, the hypophagic effect of PACAP was reduced by SHU9119, an MC3/MC4 receptor antagonist, which had no significant effect on food consumption [37]. These authors concluded that central administration of PACAP exerts its anorexic effect, at least in part, through activation of the hypothalamic melanocortin system [37].

Activation of POMC neurons suppresses food intake and appetite via the release of α-MSH, an endogenous melanocortin receptor agonist. In contrast, AgRP inhibits POMC neurons by blocking melanocortin receptors, increasing appetite and food intake [6]. PACAP was shown to interact with POMC neurons via PAC1 and VPAC2 receptors but only PAC1 receptors with AgRP neurons [6]. Nguyen and colleagues found that PACAP knockout mice had decreased AgRP mRNA expression levels and increased POMC mRNA levels after fasting and refeeding. NPY and AgRP expression was upregulated by feeding after a fasting period in both knockout and wild-type mice. 

However, AgRP expression was higher in wild-type compared to knockout mice. POMC expression was downregulated after fasting and refeeding, with the extent of the downregulation increased in knockout compared to wild-type mice. An ICV injection of PACAP6-38 decreased body weight, food consumption, and AgRP expression after fasting and refeeding in wild-type mice but with no changes in POMC expression [6]. These results suggest that PACAP may act as a regulator of anorexigenic and orexigenic peptides in the hypothalamus. Considering that the authors measured the expression of these peptides, further studies are needed to determine if endogenous or exogenous PACAP would regulate the level of these peptides and especially the release of these peptides.

PACAP was reported to decrease NPY-stimulated food and water intake in male Sprague–Dawley rats [54]. An earlier study found that PACAP (2.5 µg, ICV) decreased food consumption and increased locomotor activity in fasted mice [11]. When mice were administered NPY, food intake increased. Furthermore, a combination of PACAP and NPY induced a decrease in food consumption and increased locomotor activity, suggesting that PACAP and NPY exert opposing effects on food intake in the brain. Despite PACAP-induced suppression of NPY-mediated food intake in rats, it did not alter the level of NPY in the hypothalamus [54]. However, further studies using lower doses of PACAP are needed to differentiate whether the effect of PACAP is due to an action of the peptide on food intake or the peptide selectively alters the NPY-stimulated food intake.

Nakata and colleagues (2004) showed that food intake was reduced in PACAP-deficient mice compared to their wild-type controls. They also examined the underlying mechanism of this regulatory action of PACAP and discovered that NPY mRNA was reduced in PACAP-deficient mice. They also showed that PACAP increased intracellular calcium ([Ca^2+^]i) in NPY-containing neurons of the ARC [16]. 

Ca^2+^ is a regulator of neuronal activities; it was shown that orexigenic substances of physiological relevance, such as low glucose, ghrelin, and orexin, induce increases in [Ca^2+^]i in NPY-containing neurons [16]. Given that NPY neurons are affected by glucose and other chemicals, these authors also studied if consumption of a high-carbohydrate diet would be changed in these mice. Their results revealed that consumption of a high-carbohydrate but not high-fat diet was reduced in PACAP knockout than wild-type control mice [16]. Together, these findings suggest that endogenous PACAP influences carbohydrate intake via the NPY-releasing neurons in the ARC.

Per Shioda et al. [55], PACAP can increase or decrease the activity of glutamate ionotropic and metabotropic receptors [56] by activating Src protein tyrosine kinases, suggesting that PACAP regulates caloric intake by activating or inhibiting satiety circuits, like in the VMN; and appetite signals, in the NAc; suppressing homeostatic and hedonic feeding drives, which can be helpful towards developing a treatment for obesity and overeating.

AP5, an NMDA receptor antagonist injected into the VMN of male Sprague–Dawley rats at doses ranging from 10 to 1 nmol, did not seem to alter the feeding behaviors but blocked the effects of PACAP injected into the VMN on food intake when administered before PACAP injections [57]. Appetite and food intake were also decreased by inhibiting ghrelin and stimulating GLP1 and leptin [15]. Per Adams et al. [58], PACAP’s role is to maintain the energy balance. PACAP-null mice were not obese or hyperphagic compared to wild-type mice regardless of the temperature or diet. PACAP-null mice were initially leaner than wild-type mice if maintained on a regular chow diet and at 21 °C. 

Researchers proposed that the decrease in body mass could not be attributed to hypophagia, increased metabolism, or motor activity but rather sensitivity to cold temperatures in these mice. The reduced weight was due to decreased fat mass, leptin, and insulin levels. Researchers found that PACAP-null mice compared with wild-type mice had an improved response to insulin tolerance but normal glucose tolerance. Furthermore, adult PACAP-null mice were sensitive to cold at 4 °C, as they lost temperature faster than wild-type mice, suggesting that the thyroid axis is affected because of an observed reduction in thyrotropin-releasing hormone (TRH) mRNA. The study concluded that the reduced body mass of PACAP-null mice could be reversed with a thermoneutral environment or a high-fat diet.

In chicks, PACAP and CRH both inhibited food intake and induced behavioral changes after central injection, suggesting that the anorexigenic effects may be regulated by CRH pathways [59]. In the goldfish, studies on the effect of CRH on feeding behavior after central injection showed that it acts as a hypothalamic anorexigenic peptide [60]. PACAP was shown to regulate the stress response and affective aspect of pain via hypothalamic and extra-hypothalamic structures. For example, PACAP infusions into the central CeA caused pain hypersensitivity and increased anxiety in an elevated plus-maze [61]. Similarly, PACAP infusions in the BNST heightened acoustic startle [20], increased peripheral corticosterone release [62], caused an anorexic phenotype [47], and reinstated cocaine-seeking behavior [63].

The anorexigenic effects of PACAP and VIP in male layer chicks were mediated by CRH, as this action of the peptides was reduced by astressin, a CRH receptor antagonist [64]. However, the CRH antagonist had a differential effect on the anorexigenic effect of PACAP and VIP. Astressin significantly reversed the anorexigenic effects of PACAP at 90 min with lesser effect at earlier time points. In contrast, it significantly reversed the hypophagic effect of VIP throughout the 90-min test period, suggesting CRH is differentially involved in the hypophagic effect of VIP and PACAP. Both peptides also increased corticosterone secretion via the CRH system. However, VIP-induced corticosterone secretion was only partially attenuated by astressin [64].

The endogenous PACAP/PAC1 receptor system may also be involved in the regulatory action of leptin [32]. Animals injected with leptin into the VMN ate less than controls for 3 h after injection, and this effect was blocked by PACAP6–38. Administration of PACAP6–38 alone in the VMN had no effects on food intake or body weight. Leptin administration also increased body temperature 20 h after injection. This response was also reversed by PACAP6-38, injected before leptin in the VMN. These results suggest leptin may exert some of its action via the PAC1R activation in the VMN.

Researchers found that PACAP mRNA was co-localized with brain-derived neurotrophic factor (BDNF) mRNA and that PAC1R mRNA was co-expressed with BDNF and the leptin receptor [32]. When PACAP was administered in the VMN, researchers observed STAT3 phosphorylation and increased BDNF and SOCS3 mRNA levels, indicators of leptin receptor activation. These effects of PACAP and leptin were blocked by PACAP6-38, suggesting that PACAP is downstream of leptin and mediating its action on food intake. We propose that there may be a positive feedback loop, where central PACAP release causes leptin release from the periphery, and leptin, in turn, activates the SF-1/PACAP expressing neuron in the VMN resulting in excitation of POMC neurons [26,65].

Bath application of PACAP increased neuronal activity, although administration of leptin had a bidirectional effect in which cells that generally fired less often showed an increase in firing rate. In contrast, cells that fired more often showed a decrease in firing rate after leptin administration. Administration of PACAP6–38 attenuated the effects of leptin on firing and entirely blocked the stimulatory effects of PACAP on neuronal activation.

In summary, these findings suggest that PACAP exerts its anorectic effects via different peptide systems, including α-MSH, NPY, AgRP, leptin, and CRH. However, the involvement of different peptides has been shown in different species. Thus, although the anorectic effect of PACAP is conserved in several species, further studies are needed to delineate the role of orexigenic and anorexigenic peptides in the action of PACAP in the brain and the periphery. It is important to assess if melanocortin receptor antagonists would alter the action of PACAP in different species to examine if the role of α-MSH is also conserved in different species. Importantly, most of the studies have measured changes in the expression of orexigenic and anorexigenic peptides. It is better to measure the release of these peptides in response to PACAP administration.

## 6. The Role of PACAP Receptor(s) in the Regulatory Actions of PACAP on Feeding

PACAP activates PAC1 receptors with higher affinity than VIP. On the other hand, PACAP and VIP exhibit similar affinity toward VPAC1 and VPAC2 receptors. The PACAP and PAC1 system is thought to play a key role in feeding and energy homeostasis in the hypothalamus [66]. Still, few studies assessed the role of extrahypothalamic PACAP in these processes. PACAP was shown to increase POMC and MC4R mRNA levels in the ARC. The melanocortin system was reported to mediate the central effects of PACAP on thermogenic sympathetic and digestive parasympathetic outflow [67]. Based on this finding, it is speculated that PAC1 receptors are found presynaptically on POMC neuronal terminals to increase the synthesis of α-MSH.

As stated above, ICV PACAP6-38 injection did not affect food intake in mice deprived of food for 18 h but attenuated the inhibitory effect of PACAP on feeding during the 60-min observation period [37]. It was also found that the PACAP antagonist, PACAP6-38, attenuated the inhibitory effects of PACAP on feeding, a response that was not mimicked by VIP, indicating that PACAP has an anorexigenic action through the PAC1 receptor. 

Another study [16] concluded that PACAP increased depolarization of NPY neurons of the ARC via PAC1 and VPAC2R, resulting in reduced carbohydrate intake by promoting glucose output [43], which may prevent hypoglycemia and lack of energy in PACAP deficient mice. These findings suggest that PACAP has dual insulinotropic and glucagonotropic effects of regulating energy and glucose homeostasis. Given that PACAP could act not only on PAC1 but also VPAC1 and VPAC2 receptors and that PACAP6-38 could activate VPAC2 receptors [68], further studies are needed to characterize the receptor type involved in this process.

## 7. Signaling Involved in the Modulatory Actions of PACAP on Food Intake

According to a study by Dore and colleagues, PACAP released after stress exposure induces CRH expression via the activation of PAC1 receptors in the PVN and increases intracellular cAMP/protein kinase A signaling [66]. Therefore, CRH can be a target and mediator of PACAP’s actions. However, the weight loss and anorectic effects of PACAP seem to be independent of CRH. Researchers found that the anorexigenic effects of PACAP were not blocked by the CRH receptor antagonist D-Phe-CRF(12–41) but attenuated by SHU9119, a melanocortin receptor (MCR3/MCR4) antagonist, and by a tyrosine kinase B (TrKB) inhibitor k-252. These observations suggest that the anorexigenic effects of PACAP may be mediated by the melanocortin and TrkB/BNDF pathways [50].

Tyrosine phosphorylation of the N-methyl-D-aspartate receptors (NMDA) receptor by PACAP is associated with increased Src family kinase activity [69,70]. In particular, PACAP induces tyrosine phosphorylation of the GluN2B subunit in the NMDA receptor through Src kinase activation, which plays an important role in food intake regulation. This view was based on the finding that VMN PACAP injections produced hypophagia and GluN2B tyrosine phosphorylation [57]. The state of phosphorylation of NMDA by Src kinases is also essential in this regard, as protein phosphatase-1 (PP1), which inhibits the tyrosine phosphorylation of the NMDA receptor, reduced the hypophagic effect of PACAP. 

PACAP signaling may also regulate α-amino-3-hydroxy-5-methyl-4-isoxazolepropionic acid (AMPA) receptors, as PAC1 receptor-dependent potentiation of AMPA receptors at low PACAP concentrations and VPAC2R depression of AMPA receptors at high concentrations of PACAP in the hippocampus [71]. The PACAP-PAC1 receptor signaling increases due to the AMPA receptor activity in the amygdala.

Further works are needed in this area to delineate the role of other signaling mechanisms in the hypophagic effects of PACAP. Particularly, PACAP can activate protein kinase A and protein kinase C. Thus, the role of downstream phosphorylated products of these enzymes needs to be assessed in the hypophagic effects of PACAP.

## 8. The Action of PACAP on Homeostatic and Hedonic Aspects of Feeding

Homeostatic food intake is driven by hunger, whereas hedonic food intake is driven by palatability. The lateral hypothalamic area (LHA), containing receptors for various neurohormones and neurotransmitters including GABA, TRH, orexin, and hypocretin, proves to be an area in which the homeostatic and hedonic pathways interact. The homeostatic food intake is mediated by ARC and PVH neurons, whereas hedonic food intake is mediated by the mesolimbic dopaminergic (DAergic) neurons. 

Researchers found that dopaminergic signaling within the basal ganglia attenuates homeostatically driven food intake in the LHA. Researchers found that activation of AgRP neurons in the LHA, which is thought to co-release GABA, and inhibitory projections from the BNST to the LHA prove to drive hedonic food intake. The NAc may also play a role in hedonic food intake, as it was shown to have efferent and afferent projections to the LHA, ARC, and VTA.

A previous study [32] examined the effect of PACAP on the hedonic and homeostatic aspects of feeding. These authors developed a binge-eating paradigm where animals consumed a large amount of food within a short time. Using this model of binge eating, these authors studied the neuroanatomical site of action of PACAP in regulating homeostatic and hedonic drives of food intake. It was found that injection of PACAP into the VMN decreased hunger-mediated food consumption, while injection of the peptide into the NAc did not influence this response. In contrast, local injection of PACAP into the NAc decreased the intake of high-carbohydrate and high-fat foods. This study suggests that PACAP within the VMN reduces the homeostatic aspect of feeding, while injection within the NAc suppresses the hedonic aspect of eating.

Researchers examined whether PACAP excites POMC neurons via PAC1 receptor mediation and TRPC channel activation, a potential mechanism through which PACAP could regulate the homeostatic aspects of food intake. In recordings from POMC neurons, PACAP induced depolarization and increased firing in the voltage clamp. These postsynaptic actions were abolished by PAC1 receptor antagonists, suggesting that PAC1R is involved in these actions of PACAP. They also showed that this process involved TRPC channels, phospholipase C, phosphatidylinositol-3-kinase, and protein kinase C. 

The stimulatory effect of PACAP on POMC neurons is potentiated by estradiol and attenuated under diet-induced obesity/insulin resistance [65]. PACAP normally inhibits food intake via a negative feedback loop. However, it has been proposed that, in an addictive state, PACAP acts through a positive feedback mechanism to stimulate food intake and promote drug and food addiction cycles (for a review, see [27]). The limbic system may play a functional role in the actions of PACAP on food intake since PACAP is stimulated by chronic stress in the BNST and PVN. 

Chronic stress can lead to anxious and depressive-like moods that trigger the hedonic mechanism of food intake and promote addictive cycles [27]. However, more recent work from Wagner’s research team showed that PACAP nerve terminals were optogenetically stimulated in the VMN, and Channel Rhodopsin-2 (Chr2) expression was found in the cell bodies of PACAP neurons localized in the VMN and terminals of PACAP neurons in the VTA [72]. Optogenetic stimulation and patch-clamp recordings of this PACAP pathway showed that PACAP neurons in the VMN inhibit A10 dopamine neurons within the VTA. 

Consistent with this, they showed wild-type male mice fed on an intermittent HFD showed increased anticipatory locomotion, caloric intake, heat production, meal size, bout duration, and O2 consumption. PACAP injected into the VTA (30 pmol; 0.2 L) reduced these binge-eating behaviors in males. Although binge-eating caused increased anticipatory locomotion, caloric intake, and O2 consumption in ovariectomized female mice, PACAP failed to alter these effects in these mice. 

The intake of regular chow was not altered in mice of either sex. Together, these results suggest that the effect of PACAP on hedonic feeding may be reduced in obese/diabetic and female subjects. However, further studies are needed to recapitulate these results and provide a mechanistic explanation for these changes in the action of PACAP in these subjects.

## 9. Concluding Remarks

A growing body of evidence suggests that PACAP regulates food intake in several species, including fishes, chicks, and rodents. This action of PACAP is primarily via the PACAP-preferring (PAC1) receptor, as the action of PACAP is absent in mice lacking PAC1 receptors or in rodents treated with the PAC1 receptor antagonist PACAP6-38. The anorexic effect of PACAP involves different brain regions with a selective action of the peptide on the homeostatic aspect of food intake into the VMN in the hypothalamus and NAc as the neuroanatomical site mediating the effect of PACAP on the hedonic aspect of food intake. 

However, further research is needed to define the role of other brain regions, such as the VTA, PVT, and NTS, in the regulatory action of PACAP on different aspects of food intake. Additionally, the underlying mechanism of the effect of PACAP in suppressing food intake, particularly its actions on affecting the expression and release of orexigenic and anorexigenic peptides in the hypothalamus, needs to be determined. PACAP appears to be a potential target to develop medications to alter energy and glucose metabolism and possibly binge eating. However, developing medications with selective actions on energy homeostasis without affecting the stress response and cardiovascular and other systems requires further research.

## Figures and Tables

**Figure 1 brainsci-12-00013-f001:**
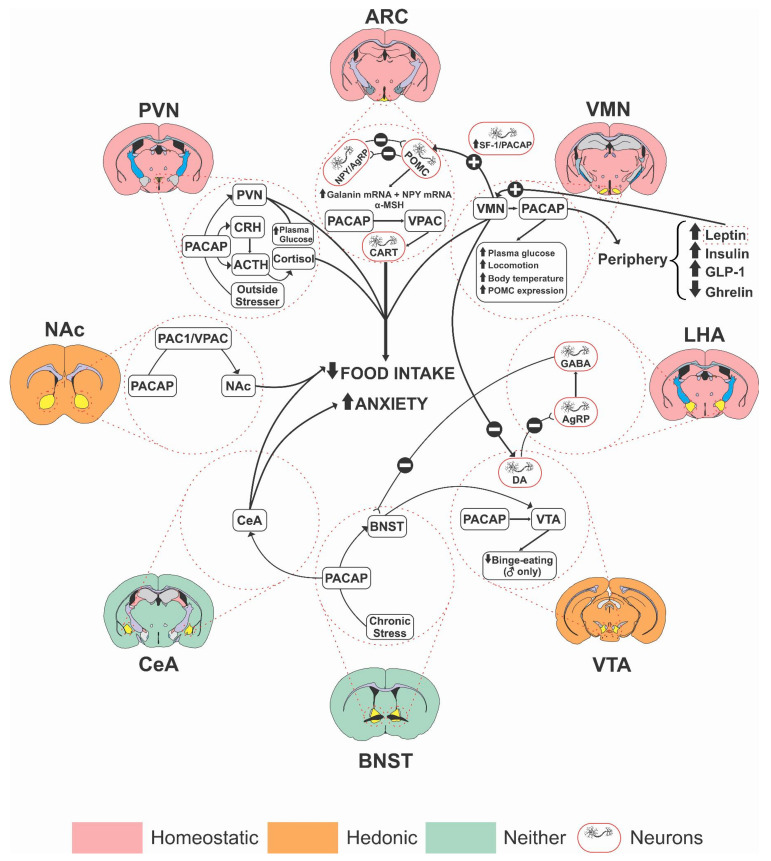
Schematic diagram of different brain regions involved in the regulatory action of PACAP on feeding. Arcuate Nucleus (ARC); Bed Nucleus of Stria Terminalis (BNST); Central Nucleus of Amygdala (CeA); Lateral Hypothalamus (LHA); Nucleus Accumbens (NAc); Paraventricular Nucleus of the Hypothalamus (PVN): Ventromedial Nucleus of the Hypothalamus (VMN); and Ventral Tegmental Area (VTA).

**Table 1 brainsci-12-00013-t001:** Summary of all the studies included in this review with subjects used, drug dose and route of administration, and their findings.

Study By	Subject(s)	Drug Dose Route	Category	Brain Area/Peptide/Peptide Precursors, etc.	Findings
**Introduction**					
Tachibana et al. (2003)	Male Chicks	ICV	Effect of various peptides on food intake	VIP, PACAP, GRF, GLP-1	VIP, PACAP, GRF, and GLP-1 decreased food intake in chicks, but VIP and PACAP with lesser potency.
**PACAP and Feeding**					
Matsuda et al. (2006)	N/A	N/A	Effect of PACAP and VIP in feeding	VMN, ARC, PVN, PACAP, VIP, NPY, AgRP, CRH, CART, leptin, orexin	Looks at the role of PACAP and VIP in feeding in vertebrates., as well as the role of orexin, NPY, AgRP, melanin-concentrating hormone, proopiomelanocortin-derived peptides, galanin, galanin-like peptide, ghrelin, CRH, cholecystokinin, CART, and leptin.
Matsuda et al. (2005)	Unspecified goldfish	IP VIP, ICV VIP, IP PACAP, ICV PACAP	Effect of PACAP and VIP in feeding	PACAP, VIP	ICV and IP PACAP, and VIP reduced food intake in the goldfish.
Mizuno et al. (1998)	Male Sprague–Dawley rats	ICV	PACAP and involvement of hypothalamic neuropeptide gene expression	Hypothalamus	The hypophagic effect of PACAP is not mediated by NPY or CRH.
Nakamachi et al. (2019)	Male and female adult short-fin strain zebrafish	ICV	Effect of PACAP on Anorexigenic Action	PACAP1, PACAP2	The expression levels and distribution of PACAP and PAC1R genes are different in zebrafish, but the hypophagic effect of PACAP is similar to those seen in other vertebrates.
**Endogenous PACAP**					
Burgos et al. (2013)	Male Sprague–Dawley rats	ICV PACAP, ICV CART	PACAP and CART on Food Intake	PACAP6-38, CART	PACAP6-38 reduced the ability of CART to decrease food intake and body weight (without altering motor activity).
Gargiulo et al. (2020)	N/A	N/A	Food and Drug Addiction Alters PACAP respons+es	PACAP27, PACAP38, PAC1 R, VIP, VPAC1, VPAC2	Review of the literature on the anorectic effects of PACAP and about the dysregulation of PACAP and its receptors following palatable food and addictive drugs.
Gray et al. (2001)	Male PAC1R (−/−) and (+/+) mice	IV Glucose (ITT)	Role of PACAP in lipid and carbohydrate metabolism	PACAP, insulin	PACAP regulates carbohydrate and fat metabolism.
Jozsa et al. (2006)	Male Wistar rats and broiler chickens	N/A	Measure levels of PACAP and VIP in response to starvation	hypothalamus, telencephalon, brainstem	PACAP levels are increased in response to starvation in both species.
Kiss et al. (2007)	Male and female rats	N/A	Role of Endogenous PACAP in food and water deprivation in various brain areas	PACAP38/hypothalamus, brainstem, and telencephalon	PACAP levels were changed in a sexually dimorphic manner in the hypothalamus, brainstem, and telencephalon in mice after food deprivation.
Sherwood et al. (2007)	N/A	N/A	Effect of PACAP on reproduction, energy homeostasis, and brain development	PACAP, VPAC, insulin, glucagon	PACAP is crucial for zebrafish brain development and reproduction, but not for body mass or food intake in mice maintained near thermoneutrality.
Hannibal et al. (2016)	Male and female PAC1R (−/−) and (+/+) mice	N/A	Role of food in anticipatory activity	PACAP	PAC1 KO mice showed an earlier onset of anticipatory activity than WT at the 10-lux light intensity. Both KO and WT had greater FAA in the SPP 10 and 300 lux, and FPP 300 lux light intensities than FPP 300 intensity.
Nakata et al. (2004)	Male PACAP (−/−) & (+/+) and rats	Superfusion	Effect of PACAP on carb intake and NPY in the ARC	PACAP, VIP, NPY, Maxadillian	NPY mRNA was reduced in PACAP-deficient mice. Carbohydrate but not fat intake was reduced in PACAP deficient mice. NPY neuron is not the sole target for PACAP’s actions in the ARC.
**Exogenous PACAP**					
Green et al. (2006)	Unspecified ob/ob mice	IV PACAP27, IV PACAP6-38, IP PACAP27, IP PACAP6-38, IP Glucose, IP Insulin	Role of endogenous PACAP in glucose and insulin levels in ob/ob mice	PACAP27, PACAP38, PACAP6-38, insulin	PACAP is involved in glucose homeostasis but less in insulin release in ob/ob mice.
Lugo et al. (2010)	Unspecified African catfish, Nile tilapia	IP PACAP, IP VIP	Role of GH on PACAP and VIP effects on food intake	PACAP, VIP, GH	PACAP increased food consumption in juvenile tilapia and may be involved in growth.
Morley et al. (1992)	Male SW mice	ICV	Effect of PACAP on food intake	PACAP, NPY	PACAP reduced basal and NPY-stimulated food intake.
Mounien et al. (2006)	Male Swiss albinos CD1 and C57BL/6	Central injection	Expression of PACAP by NPY in ARC	PACAP, NPY, PAC1R, VPACR	PACAP may regulate the activity of NPY, and the inhibitory effect of PACAP on food intake may be affected by NPY neurotransmission.
Mounien et al. (2009)	Unspecified Rat	ICV	Role of PACAP through Melanocortin system	VIP, PACAP, SHU9119	A high proportion of POMC neurons express PACAP receptors. PACAP may inhibit food intake through a melanocortin-dependent pathway.
Yokota et al. (1995)	Male Wistar-lmamichi rat	Infusion	Effect of PACAP on glucose	PACAP, theophylline, VIP, glucagon	PACAP stimulates glucose output from the perfused rat liver, and Ca^2+^ rather than cAMP plays an important role in this action.
Yada et al. (1994)	Unspecified Rat	Not Specified	Effect of PACAP on Insulin Secretion	PACAP, VIP, insulin	PACAP increases insulin release from rat islets in a glucose-dependent manner. VIP is less potent.
Yada et al. (2000)	Female GK rats and C57B/6J mice	IP	PACAP effect in blood glucose in GK rats (diabetic) and normal high fat diet mice (glucose intolerant)	PACAP, insulin	PACAP reduced hyperglycemia in GK rats and mice fed an HFD.
Yu et al. (2008)	Male NIH Mice	IP	Effect of chronic maxadilan on glucose levels and insulin sensitivity	M65, maxadilan, insulin	Maxadilan-induced hyperglycemia was blocked by M65. Acute maxadilan decreased feeding. Chronic maxadilan increased body weight and decreased body fat. PACAP increased glucose tolerance and insulin sensitivity.
Vu et al. (2015)	PAC1R (−/−) and (+/+)	IP	Effect of PACAP on other neurotransmitters to decrease food intake	PACAP, ghrelin, GLP-1, leptin, insulin, glucagon, PYY	PACAP decreased food intake and ghrelin in wild-type but not KO mice. Postprandial levels of active ghrelin and insulin were different in fasted KO than WT mice.
Tachibana et al. (2004)	Male Chicks	ICV	Effect of CRH on PACAP and VIP in chicks	PACAP, VIP, astressin, PACAP6-38	CRF neurons modulate PACAP- and VIP-induced hypophagia.
**Sites of PACAP action**					
Resch et al. (2011)	Male Sprague–Dawley rats	Microinjections	Effect of PACAP in VMN on food intake and thermogenesis	PACAP, VMN	PACAP input to the VMN can influence energy homeostasis.
Resch et al. (2013)	Male Sprague–Dawley rats	Microinjections	PACAP regulates feeding	VMN, PVN, PACAP. VIP 6-28, PACAP6-38	PACAP injected in the PVN decreased meal size, duration of the meal, and total time spent eating. PACAP in both the PVN and VMN increased plasma glucose but not pancreatic hormone levels. PACAP in the VMN increased core body temperature and spontaneous locomotor activity.
Kocho-Schellenberg et al. (2014)	Male and female Sprague–Dawley rats	ICV injection of 1.0 μg/0.25 μL PACAP38	PACAP BNST localization on food/water intake	PACAP38, BNST	Infusion of PACAP38 in the posterior (but not anterior) BNST rats induced hypophagic effects.
Iemolo et al. (2015)	Male Wistar rats	PACAP 0–1 ug/rat in the CeA and BlA	Role of PACAP in response to stress	CeA/PACAP	PACAP in the CeA exerts anorectic effects via melanocortin and TrKB.
Hurley, M.M., et al. (2016)	Male Sprague–Dawley rats	PACAP 50 pmol/0.25 uL/side/AMPA 74.5 ng/side/Intra VMN administration	Role of PACAP in obesity	VMN/PACAP, AMPA	PACAP injection into the NAc reduces hedonic feeding, while its injection into the VMN decreases homeostatic feeding.
Liu et al. (2015)	N/A	N/A	Effects of PACAP on Hedonic Food Intake	ARC, LHA, VTA, NAc, GABA, DA, POMC, NPY, AgRP	Researchers observed the interaction between food intake and reward circuits in the NAc and VTA.
Adams et al. (2008)	Male and female PAC1 (−/−) and (+/+)	N/A	Role of endogenous PACAP	Hypothalamic nuclei	PACAP is involved in maintaining energy homeostasis, possibly via cold-sensing mechanisms.
Nguyen et al. (2020)	Male PACAP (−/−) and (+/+)	ICV	Effect on PACAP in VMH on AgRP	PACAP. VIP 6-28, PACAP6-38, AgRP, VMH	PACAP regulates feeding in VMH by modulating the expression of AgRP. Knockout mice fasted for two days had significantly lower food consumption during refeeding than PACAP (+/+) mice.
Chance et al. (1995)	Male Sprague–Dawley rats	IHT Injections of CRG, PACAP, and NPY	Effect of PACAP on NPY-dependent food intake	NPY, PACAP, hypothalamus	PACAP did not alter the hypothalamic NPY level but reduced food intake when given before NPY.
**Role of PACAP receptor(s)**					
Dore et al. (2013)	Male Wistar rats	ICV PACAP (PACAP-38) and ICV D-Phe-CRF (12–41)	PACAP and CRF signaling pathways in PVN	PACAP and CRF signaling pathways in PVN	CRF may mediate the anxiogenic and anti-rewarding effects of PACAP but not its anorectic action.
**Signaling**					
Resch et al. (2014)	Male Sprague–Dawley rats	Injection	NMDA mediates PACAP in VMH	VMN, glutamate, Src kinase, PACAP	PACAP in the VMN increases glutamate signaling through NMDA receptors and phosphorylation of the Src kinase to regulate food intake.
**Homeostatic and Hedonic Aspects**					
Le et al. (2021)	Male and female C57BL/6 mice	Local injection	Pituitary Adenylate Cyclase Activating Polypeptide Inhibits A _10_ Dopamine Neurons and Suppresses the Binge-like Consumption of Palatable Food	VMN, VTA, DA	PACAP injected into the VTA reduced hedonic food intake and dopaminergic neuronal activity.
Chang et al. (2020)	Male and female mice	IV	Effect of PACAP on POMC neurons in mediating homeostatic food intake	PACAP, CNO, E2, ETOH, STX, PACAP6-38	PACAP-stimulated POMC neurons were attenuated in mice on HFD. Elevation of PACAP in the VMN and ARC by chemogenetic and optogenetic stimulation decreased energy intake and increased energy expenditure.
Chang et al. (2019)	Male and female eGFP-POMC mice, PACAP-Cre mice	Local injection	Effect of PACAP on POMC neurons in mediating homeostatic food intake	POMC, TRPC channels, VMN, ARC	PACAP activates POMC neurons via the PAC1 R involving TRPC5 channels at the VMN and ARC, mediated by estradiol, although these effects are not seen in HFD mice.

## Data Availability

Not applicable.

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
