# Peer review of "The PACAP/PAC1 Receptor System and Feeding"

_brainsci, 2021, doi:10.3390/brainsci12010013_

Round 1

Reviewer 1 Report

Few changes to the text are required
Pag. 3 in The "Findings" of Bernier NJ, Bedard N, Peter Re (2004)  the point after "diet group"is missing. 
Pag. 4 In The "Findings" of Burgos Jr, Iresjö BM, Smedh U (2013) the point after parenthesis is missing.
Pag. 4 In The "Findings" of Burgos Jr, Iresjö BM, Smedh U (2013) there is the comma instead of the point after "Activity".
It is necessary to check out all the work to adjust the punctuation.
Page 15 L 78-81: Better eliminate this paragraph, it is quite useless to say what is not the purpose of the review.
Page 18 L 228: You need to specify that the knockouts is for the PAC1 receptor (see Hannibal et al. 2016) 

Author Response

We thank the reviewer for the critical review of our manuscript and the comments provided, which enhanced the quality of our manuscript. We have incorporated all the comments of the reviewer in our revised manuscript.
Our responses to each comment are as follows:
Pag. 3 in The "Findings" of Bernier NJ, Bedard N, Peter Re (2004)  the point after "diet group"is missing. 
Response: According to Reviewer 2, we have modified the finding section of Table1. It may not be applicable anymore. If it is applicable, we have modified it.
Pag. 4 In The "Findings" of Burgos Jr, Iresjö BM, Smedh U (2013) the point after parenthesis is missing.
Response: As stated above, we have modified the finding section of Table1. It may not be applicable anymore. If it is applicable, we have modified it.
Pag. 4 In The "Findings" of Burgos Jr, Iresjö BM, Smedh U (2013) there is the comma instead of the point after "Activity".
Response: As stated above, we have modified the finding section of Table1. It may not be applicable anymore. If it is applicable, we have modified it.
It is necessary to check out all the work to adjust the punctuation.
Response: We have modified the manuscript as much as possible. However, if the reviewer has any specific suggestion(s), we will be happy to address them.
Page 15 L 78-81: Better eliminate this paragraph, it is quite useless to say what is not the purpose of the review.
Response: As suggested, we have eliminated these sentences.
Page 18 L 228: You need to specify that the knockouts is for the PAC1 receptor (see Hannibal et al. 2016)
Response: We have incorporated PAC1 knockout in the text of our revised manuscript.

Reviewer 2 Report

The manuscript entitled ”The PACAP/PAC1 receptor system and feeding” is a review of 68 articles about the regulatory effect of PACAP and PAC1 receptor on food intake. The manuscript summarizes the physiological roles of PACAP and its receptors, the role of endogenous and exogenous PACAP and its receptors in food intake, the neuroanatomical sites of PACAP action in regulating food intake, the effect of other neurotransmitters on PACAP.
The review is really detailed and summarizes the most important articles of this field, but the interpretation needs further improvement.
Main remarks:
1., The Table 1 is really detailed, it is almost the repetition of the whole manuscript, it is recommended to shorten the finding column of the table to only a few words.
2., In  Table 1. the citations are in alphabetic order. It is recommended to form smaller separated parts based on the parapgraphs and follow the structure of the manuscript.
3., The individual paragraphs contain the summary of the different manuscripts, but there are no connections between them. First the authors write about chicken, after mice, after KO animals and goldfish. The structure is not logical. It is recommended to put the different results in logical order. (From fish to mice for example).
4., The results are mixed within different paragraphs, we can read about the effect of exogenous PACAP in the paragraph about the role of endogenous PACAP and we can find results about endogenous PACAP in the part about the effect of different PACAP treatments.
5., The detailed description about PACAP 6-38, 6-27 is missing from the manuscript. What are those?  What are the main general functions?
6., Figure 1 is a very good summary, but unfortunately it is too small to read.
7., It is recommended to insert more Figure to the different paragraphs similarly to Figure 1. (for example about the neurotransmitters).
This review is a good summary about the effect of PACAP and its receptors in the food intake. But further improvement is necessary to publish in this journal.  

Author Response
We thank the reviewer for the critical review of our manuscript and the inshightful comments provided, which enhanced the quality of our manuscript. We have incorporated all the comments of the reviewer in our revised manuscript.
Our responses to each comment are as follows:
Main remarks:
1., The Table 1 is really detailed, it is almost the repetition of the whole manuscript, it is recommended to shorten the finding column of the table to only a few words.
Response: As suggested, we have modified these items of Table 1.
2., In  Table 1. the citations are in alphabetic order. It is recommended to form smaller separated parts based on the parapgraphs and follow the structure of the manuscript.
Response: As suggested, we have reordered the content of the table. Now, the content of Table 1 and the order of the presentation of studies in the manuscript is consistent.
3., The individual paragraphs contain the summary of the different manuscripts, but there are no connections between them. First the authors write about chicken, after mice, after KO animals and goldfish. The structure is not logical. It is recommended to put the different results in logical order. (From fish to mice for example).
Response: Our goal was to give credit to the authors based on the chronological order of the studies, which were conducted in rodents in early 1990s and then studies in other species.followed these.
4., The results are mixed within different paragraphs, we can read about the effect of exogenous PACAP in the paragraph about the role of endogenous PACAP and we can find results about endogenous PACAP in the part about the effect of different PACAP treatments.
Response: We apologize for this oversight. According to your suggestion, we have modified these sections to address this issue. At the end of each section, the highlighted in yellow, are summaries of each section or connecting the dots.
5., The detailed description about PACAP 6-38, 6-27 is missing from the manuscript. What are those?  What are the main general functions?
Response: In the studies we reviewed, we have provided bried description of PACAP6-27 and PACAP6-38 in  our revised manuscript.
6., Figure 1 is a very good summary, but unfortunately it is too small to read.
Response: We have enlarged the fonts in figure 1 to be more visible.
7., It is recommended to insert more Figure to the different paragraphs similarly to Figure 1. (for example about the neurotransmitters).
Response: Some of the information of the neurotransmitters are provided in Figure 1. Given the Editor gave us five days to submit the revised manuscript, we were not able to create more figures.
This review is a good summary about the effect of PACAP and its receptors in the food intake. But further improvement is necessary to publish in this journal.  
Response: As stated above, we have modified the text and interpreted the result as much as possible (please see the highlighted section at the end of each section).